# A novel comprehensive metric to assess effectiveness of COVID-19 testing: Inter-country comparison and association with geography, government, and policy response

**Anthony C. Kuster**[1], **Hans J. Overgaard**[2,3]*

**1** Faculty of Public Health, Khon Kaen University, Khon Kaen, Thailand, **2** Faculty of Science and Technology, Norwegian University of Life Sciences, Ås, Norway, **3** Department of Microbiology, Faculty of Medicine, Khon Kaen University, Khon Kaen, Thailand

* hans.overgaard@nmbu.no

**Data Availability Statement:** The full dataset that was analyzed in this paper has been uploaded as a csv. file. Our data was obtained from publicly available data, including: Worldometer COVID-19

## Abstract

Testing and case identification are key strategies in controlling the COVID-19 pandemic. Contact tracing and isolation are only possible if cases have been identified. The effectiveness of testing should be assessed, but a single comprehensive metric is not available to assess testing effectiveness, and no timely estimates of case detection rate are available globally, making inter-country comparisons difficult. The purpose of this paper was to propose a single, comprehensive metric, called the COVID-19 Testing Index (CovTI) scaled from 0 to 100, derived from epidemiological indicators of testing, and to identify factors associated with this outcome. The index was based on case-fatality rate, test positivity rate, active cases, and an estimate of the detection rate. It used parsimonious modeling to estimate the true total number of COVID-19 cases based on deaths, testing, health system capacity, and government transparency. Publicly reported data from 165 countries and territories that had reported at least 100 confirmed cases by June 3, 2020 were included in the index. Estimates of detection rates aligned satisfactorily with previous estimates in literature ($R^2$ = 0.44). As of June 3, 2020, the states with the highest CovTI included Hong Kong (93.7), Australia (93.5), Iceland (91.8), Cambodia (91.3), New Zealand (90.6), Vietnam (90.2), and Taiwan (89.9). Bivariate analyses showed the mean CovTI in countries with open public testing policies (66.9, 95% CI 61.0–72.8) was significantly higher than in countries with no testing policy (29.7, 95% CI 17.6–41.9) (p<0.0001). A multiple linear regression model assessed the association of independent grouping variables with CovTI. Open public testing and extensive contact tracing were shown to significantly increase CovTI, after adjusting for extrinsic factors, including geographic isolation and centralized forms of government. The correlation of testing and contact tracing policies with improved outcomes demonstrates the validity of this model to assess testing effectiveness and also suggests these policies were effective at improving health outcomes. This tool can be combined with other databases to identify other factors or may be useful as a standalone tool to help inform policymakers.

(April 20, 2020 and June 3, 2020) (https://www.worldometers.info/coronavirus/) provided data on Total Cases, Total Deaths, Total Recovered cases, Active Cases, Total Tests and population. The Economist Intelligence Unit Democracy Index (2019) (https://www.eiu.com/topic/democracy-index) provided data for the Democracy Index. Global Health Security Index (October 2019) (https://www.ghsindex.org/report-model/) provided data for the Detection and Reporting Country Score. Oxford Coronavirus Government Response Tracker (May 13, 2020) (https://ourworldindata.org/grapher/covid-19-testing-policy, https://ourworldindata.org/grapher/covid-contact-tracing) provided data on COVID-19 Testing Policies, which countries do COVID-19 contact tracing?

**Funding:** The authors received no specific funding for this work, apart from publication costs covered by the Norwegian University of Life Sciences.

**Competing interests:** The authors have declared that no competing interests exist.

## Introduction

Coronavirus disease-2019 (COVID-19) is caused by infection of the Severe Acute Respiratory Syndrome Coronavirus 2 (SARS-CoV-2). The COVID-19 pandemic has forced many countries, states, and territories to enact public health measures to reduce its spread, including social distancing, contact tracing, stay-at-home orders, shuttering of schools, closure of public spaces, and border closures [1, 2].

Testing, case identification, and isolation are critical activities to breaking the transmission chain [3, 4]. Other measures, including social distancing and use of face masks, are also needed [5]. In order to assess testing and inform decisions about resuming economic activities, many countries and institutions have tracked testing-related metrics.

Thus far, the approach of many countries has generally been to specify several separate metrics related to testing, such as incidence, test positivity rate, number of hospitalizations, and mortality rate, with benchmarks for each criteria that are used to justify removing or reinstating measures in phases [6, 7]. However, many of these metrics rely on cases that have been identified through active diagnostic testing. One challenge, though, is the substantial proportion of the infected population that is asymptomatic [4, 8, 9]. Consequently, diagnostic testing has been inadequate to reveal what proportion of the population is infected, with real infections in most countries estimated to be 10 to 15 times, and sometimes even >100 times, higher than the reported number of cases [10–12]. Furthermore, predicating reopening dates on incidence may disincentivize testing, since increased diagnostic testing will inherently uncover more cases and, thus, delay reopening. Another criticism is that some criteria, such as the "downward trajectory," specified by the US Centers for Disease Control [6], are vague. Thus, metrics used by policymakers and politicians to inform decisions should not only be quantitative but also encourage widespread proactive testing, such as the proportion of the total number of infections that have been detected [13]. However, estimating the detection rate/underreporting is challenging.

The level of undetected cases has been estimated with models using transmission simulations and flight data [14, 15]. Generally, these approaches require location-specific inputs, limiting scalability and transferability. Alternatively, deaths seem to be a good indicator of true number of COVID-19 infections in the population [16, 17]. Deaths have been used in past pandemics to estimate the true size of the pandemic given limited case identification [18, 19]. The infection-fatality rate (IFR) of COVID-19 based on serological testing and comprehensive diagnostic testing has been shown to be between 0.7% and 1% in the early stages of the pandemic [20–23], indicating that approximately 100 infections have occurred to each death. However, factors such as health system capacity, demography, and political regime impact the IFR [24]. Furthermore, heterogeneity in definitions of COVID-19-related deaths and testing strategies cause differences in completeness of the death count [25]. Thus, if deaths are used as an indicator of the true number of infections, adjustments may be necessary for testing, health system capacity, and government transparency.

A single, comprehensive metric that is scalable across all countries and territories would allow comparisons across states and identify ones most successful in more completely detecting the presence of infections in the population. It would also allow for statistical techniques, such as multiple linear regression, so researchers could comprehensively assess policy decisions in combination with other databases. Comprehensive metrics for COVID-19 and inter-country comparisons have been developed [26–29], but none that focus exclusively on testing or explicitly incorporate the true number of infections or detection rate, to the best of the authors' knowledge. Thus, there is still a need for a single comprehensive metric that can overcome shortcomings in reported data to assess testing effectiveness during the COVID-19 pandemic.

Therefore, a single comprehensive metric that assesses testing effectiveness by incorporating an estimate of the true total number of COVID-19 infections in the population was developed using publicly reported data universally accessible across nearly all countries and territories. Model estimates of true period prevalence and detection rate were validated against comparable estimates in the literature. The metric was then used to assess factors associated with COVID-19 testing outcomes. We aimed to create a new tool for policymakers and researchers to comprehensively assess COVID-19 testing outcomes and identify effective policies.

## Methods and materials

### Data input

Data on COVID-19 were collected from the Worldometer website, which collects data directly from government communication channels and is managed by an international team of developers, researchers, and volunteers [30]. The data collected from Worldometer included total cases (C), total deaths (D), total recovered (R), active cases (A), population (P) (in millions), and total tests (T). These data have been reported explicitly or implicitly by the website since at least April 6, 2020. Prior to this date, subsets of these data were available.

Two other input data included the Global Health Security Index Detection and Reporting sub-index ($I_{sys}$) [31] and the Economist Intelligence Unit Democracy Index ($I_{dem}$) [32]. The $I_{sys}$ assesses a health system's capacity for early detection and reporting during epidemics of potential international concern. This index is available for 195 countries and has a scale from 0 to 100, where 100 indicates perfect detection and reporting. $I_{sys}$ values were not available for Hong Kong and Taiwan. The value for Hong Kong, $I_{sys} = 78$, was imputed as the average between South Korea ($I_{sys} = 92.1$) and Singapore ($I_{sys} = 64.5$), which were assumed to have comparable health systems. Similarly, the value for Taiwan, $I_{sys} = 81$, was imputed as average between South Korea ($I_{sys} = 92.1$) and Japan ($I_{sys} = 70.1$). All other states without a value (n = 10) were imputed as 42, which was the global average. $I_{dem}$ is a projected measure of the degree of democracy; it is calculated for 167 countries and states and has a scale from 0 to 10, where 10 indicates the highest degree of democracy. This value was assumed to act as a proxy for transparency in data reporting. All states without a value (n = 14) were imputed to be 5.4, the global average.

The inclusion criteria for incorporating countries and territories in the index calculations were if at least 100 cases had been reported and the population was greater than or equal to 100,000. The threshold of 100 confirmed cases was arbitrarily chosen to exclude locations where an outbreak had not yet occurred, in which case an analysis of testing policy was not relevant. Data were accessed daily; however, this report presents the results for data accessed as of 00:00 GMT on June 3, 2020 (n = 165).

### Definition of key indicators

Several key indicators were computed from the input data, representing important epidemiological indicators used in the analysis.

### Case Fatality Rate (CFR)

The CFR is the proportion of total deaths, *D*, among closed cases (sum of *D* and *R*). However, some countries, including the Netherlands and United Kingdom, have not reported the number of recoveries, and others have not tracked recoveries in real-time. Additionally, the closed-case definition of CFR can overestimate the CFR in the early stages of an epidemic because of

the relatively small number of closed cases [33]. In these situations, the CFR is either incomputable or artificially high. Therefore, an alternative estimate of CFR was necessary. The ratio of deaths, $D$, to cases, $C$, which can be computed for any country, was used. Logically, the ratio of D:C is lower than the closed-case definition of CFR, D:(D+R), because C includes unresolved cases with unknown outcomes. These two ratios are related though. A scatterplot of these ratios revealed a positive relationship ($R^2 = 0.76$, n = 157), in which the closed-cased definition of CFR was between one and two times the ratio of D:C in 68.5% of countries. However, in some countries, the closed-case definition of CFR was substantially inflated relative to D:C and deviated from the linear relationship. Two possible scenarios would likely explain this—either case resolution (count of R) was not tracked in real-time, or a recent outbreak occurred, in which a large proportion of cases were recently identified and had not yet resolved. Thus, the CFR, as computed using the reported closed-case definition, was used in our further analysis but was capped to be no greater than 2 times the ratio of *D:C* (Eq 1), in order to exclude artificially inflated CFRs.

$$CFR = \min\left(\frac{D}{D+R}, \ \frac{2D}{C}\right) \tag{1}$$

**Test Positivity Rate (TPR).**   TPR was computed as the ratio of cases, $C$, to tests, $T$ (Eq 2).

$$TPR = \frac{C}{T} \tag{2}$$

While the reported number of tests from specific countries or territories may represent multiple tests conducted on a single individual or even number of specimens, no adjustment was attempted to account for such heterogeneity in the various definitions of the TPR. In some countries, $T$ was not available and thus TPR was not calculated.

**Active Cases (AC).**   AC was computed as the ratio of active cases, $A$, to cases, $C$ (Eq 3).

$$AC = \frac{A}{C} \tag{3}$$

In some countries, $A$ was not available and thus AC was not calculated.

## Estimating true number of infections and detection rate

It can be assumed that the reported number of cases, $C$, in a country represents a subset of the true number of infections. Some infections will go undetected, but as detection of cases increases, $C$ will approach the true number of infections. Thus, the true number of infections (*Inf*) is some factor, $f$, higher than the cases that have been identified (Eq 4).

$$Inf = fC \tag{4}$$

We conceptualized this factor, $f$, to be a function of the level of testing, the approach to testing (e.g., whether testing focused on symptomatic, hospitalized, or general populations), and the quality and completeness of the data. Two variations of $f$ ($f_1$ and $f_2$) were constructed to formulate a numerical value for $f$, where $f_1$ was derived from CFR, $I_{sys}$, $I_{dem}$ and $f_2$ used TPR as described below.

**Factor 1 ($f_1$).**   Deaths have been used as an indicator of the true prevalence of an infectious disease and a means to track underreporting in real-time [19, 34]. The proportion of *Inf* that resolve to death is the infection fatality rate (IFR). In the first four months of the pandemic, the IFR for COVID-19 was around 1% or less [20]. That is, for every recorded death, at least

100 infections likely occurred. Thus, if less than 100 cases were recorded per death attributed to COVID-19 (i.e., the ratio of 100D:C exceeded 1), it was an indication of under-detection of infections. The primary factor, $f_1$, was computed in such a way to create a multiple that scales *Inf* to represent at least 100 infections occurring per attributed death (Eq 5).

$$f_1 = m_{demsys} \max\left(\frac{100D}{C}, 1\right) \qquad (5)$$

However, it was also assumed that health system capacity and government transparency affect the completeness of data, including reported deaths, as represented by the factor $m_{demsys}$ in Eq 5. The multiplier intended to adjust for the health system capacity using $I_{sys}$—an indicator of a health system's ability to detect and report deaths and/or cases; and $I_{dem}$—a proxy for government transparency in reporting data. The relationship between $I_{dem}$, $I_{sys}$, and $m_{demsys}$ (Eq 6) was determined by fitting the data to estimates of the true number of infections in 35 countries across a wide range of populations and regions where the estimate was available from an existing model that used machine learning to estimate the number of infections [35]. In short, an existing model was used to determine what the $m_{demsys}$ multiplier would have needed to be to reach that existing model's estimates and then fit the data using multiple linear regression. A linear model was chosen since it matched the relationship better than power and non-linear models.

$$m_{demsys} = 4.5 - 0.25 I_{dem} - 0.012 I_{sys} \qquad (6)$$

The mathematical relationship between the two indices and the multiplier in this equation followed a declining relationship, whereby increased health system capacity ($I_{sys}$) or increased government transparency ($I_{dem}$) reduced the multiplier, representing less underreporting.

**Factor 2 ($f_2$).** The primary factor *f* relied on deaths attributed to COVID-19, but in locations with inadequate or low testing levels, death data may also be underreported, making deaths a less representative indicator. Therefore, an alternative factor incorporated data on testing (Eq 7). This factor was based on TPR—an indicator of adequate testing relative to disease prevalence.

The World Health Organization (WHO) has suggested testing capacity is adequate when TPR is less than 5% [36]. If TPR was greater than 5%, it was inferred that increasingly more cases were undetected and that the factor $f_2$ would be equal to the ratio of the TPR to that 5% benchmark (Eq 7).

$$f_2 = \begin{cases} 1 & \text{if } TPR < 5\% \\ \dfrac{TRR}{0.05} & \text{if } \geq 5\% \end{cases} \qquad (7)$$

**True number of infections.** The two factors $f_1$ and $f_2$ provided two different ways to assess what the factor *f* might be for a given country. Generally, a high TPR is indicative of limited testing to symptomatic or severely ill patients, which will result in underestimation of infections within reported cases [17]. In such a testing strategy where only severe infections that are more likely to result in death were counted as cases, the ratio of D:C or CFR were also likely to be higher, as reflected in $f_1$. However, if D:C ratio was low but TPR was high, these two indicators were providing conflicting information. Such a case could occur early in an outbreak when only limited deaths have occurred or where deaths were not being fairly attributed as caused by COVID-19. In either case, the high TPR suggests many infections were not being detected, despite the low number of deaths relative to cases. In most cases the two factors

correlated. The factor $f_1$, based on the reliable indicator deaths, was approximately equal to or greater than the TPR factor, $f_2$. However, in cases where $f_2$, gave conflicting information to $f_1$, (i.e., $f_2 > f_1$), $f_2$ was used. Finally, the true number of infections, *Inf*, was estimated as the product of *C* and the maximum of the two factors, whichever was highest (Eq 8).

$$Inf = fC = \max(f_1, f_2)C \tag{8}$$

When possible, *Inf* was compared against other empirical country-level estimates in the literature in order to validate the model.

**Detection Rate (DR).** With an estimate of the true number of infections, it was possible to estimate what percentage of the infections (including asymptomatic) have been detected. The total number of cases, C, was divided by *Inf* to estimate the DR in each country or territory. Estimates of the DR were compared to DRs published in the literature.

## Calculation of the COVID-19 Testing Index

The COVID-19 Testing Index (CovTI) consists of four sub-indices, each based on a single key indicator. The key indicators used were the DR, TPR, CFR, and AC (Fig 1). These indicators were chosen since they are commonly used as metrics and are computable with the methods described above. The relationship between each sub-index and its indicator was built on two principles. First, each sub-index was scaled from 0 to 100 with 0 representing the worst indicator value and 100 the best indicator value. Second, each sub-index was

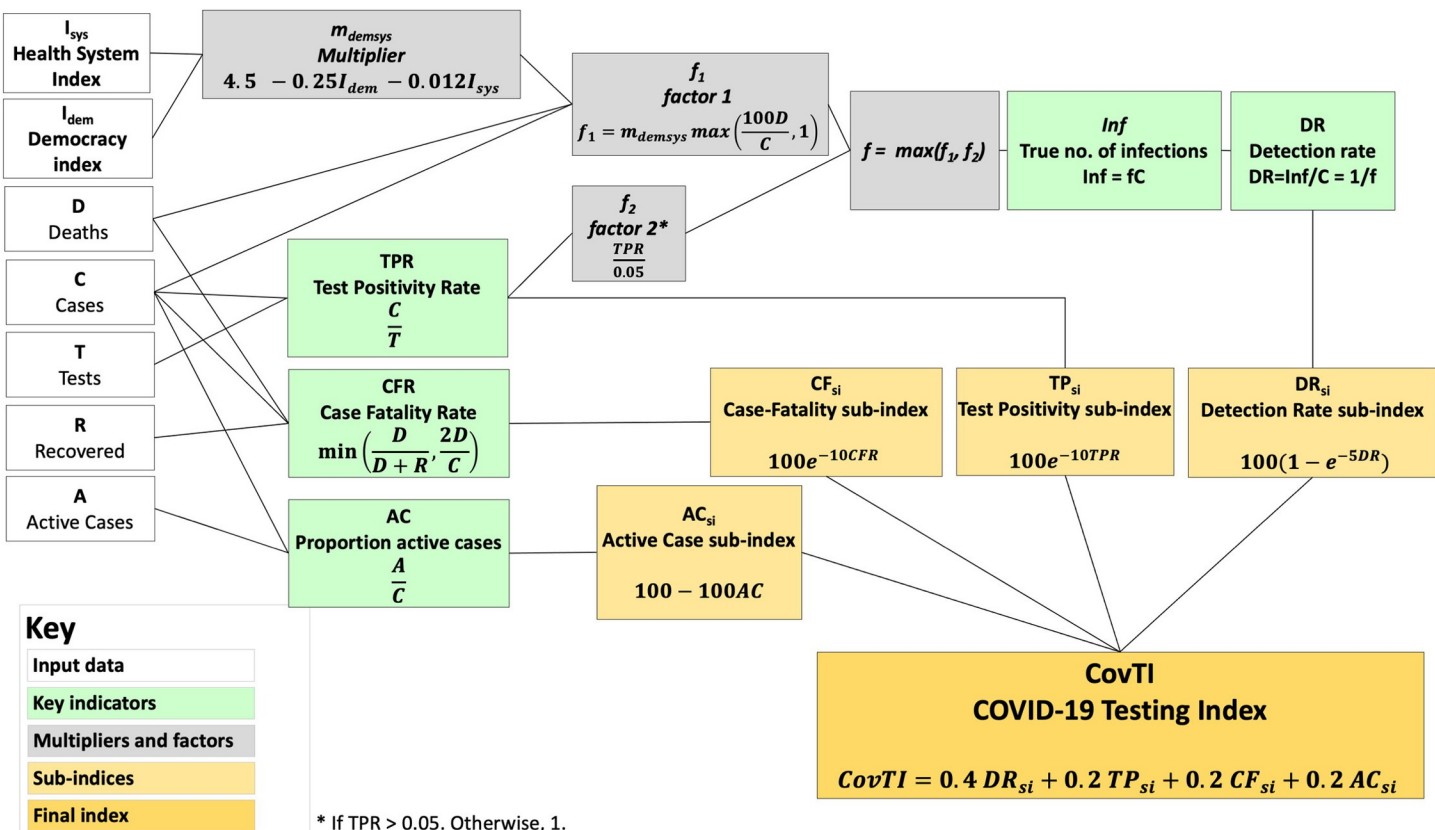

**Fig 1. Schematic diagram illustrating method to compute the COVID-19 Testing Index (CovTI).** The CovTI was computed from a weighted sum of the four sub-indices (orange), each of which is derived from one key indicator (green). The input data for each indicator (white) and intermediate steps (grey) are shown.

computed from a mathematical relationship that parsimoniously approximated the ranked percentile of the country according to the underlying indicator. Previous indices used for ecologic studies have computed the ranked percentile for the ecological units (here, countries) for use as the index value [37, 38]. We modified this approach, however, since ranking provides only relative information (e.g. how well one country is doing relative to another). We also wanted the index to have the property that an ideal value exists (e.g. 100% for DR or 0% for TPR or CFR). Thus, the countries were ranked and a function for computing the sub-index was chosen that mimicked the distribution of the countries but still retained the ability to have 100 represent the ideal value and 0 represent the worst value. These sub-indices were combined in a weighted average to compute CovTI. In the absence of a rationale to assign weighting to the sub-indices, equal weighting was used with details below [37].

**Detection Rate sub-index (DR$_{si}$).** If infections are undiagnosed, those individuals can actively spread the disease unknowingly. Without timely diagnosis, effective contact tracing cannot occur. Therefore, undiagnosed infections critically contribute to unchecked spread of COVID-19 and subsequently represent inadequate response [39]. DR is also most directly representative of testing effectiveness. Thus, DR$_{si}$ was given 40% weighting (double weighting compared to other metrics) in computing the CovTI (Eq 13). Ranking (Fig 2A) showed a relationship, in which as the detection rate increased, percentile increased asymptotically towards

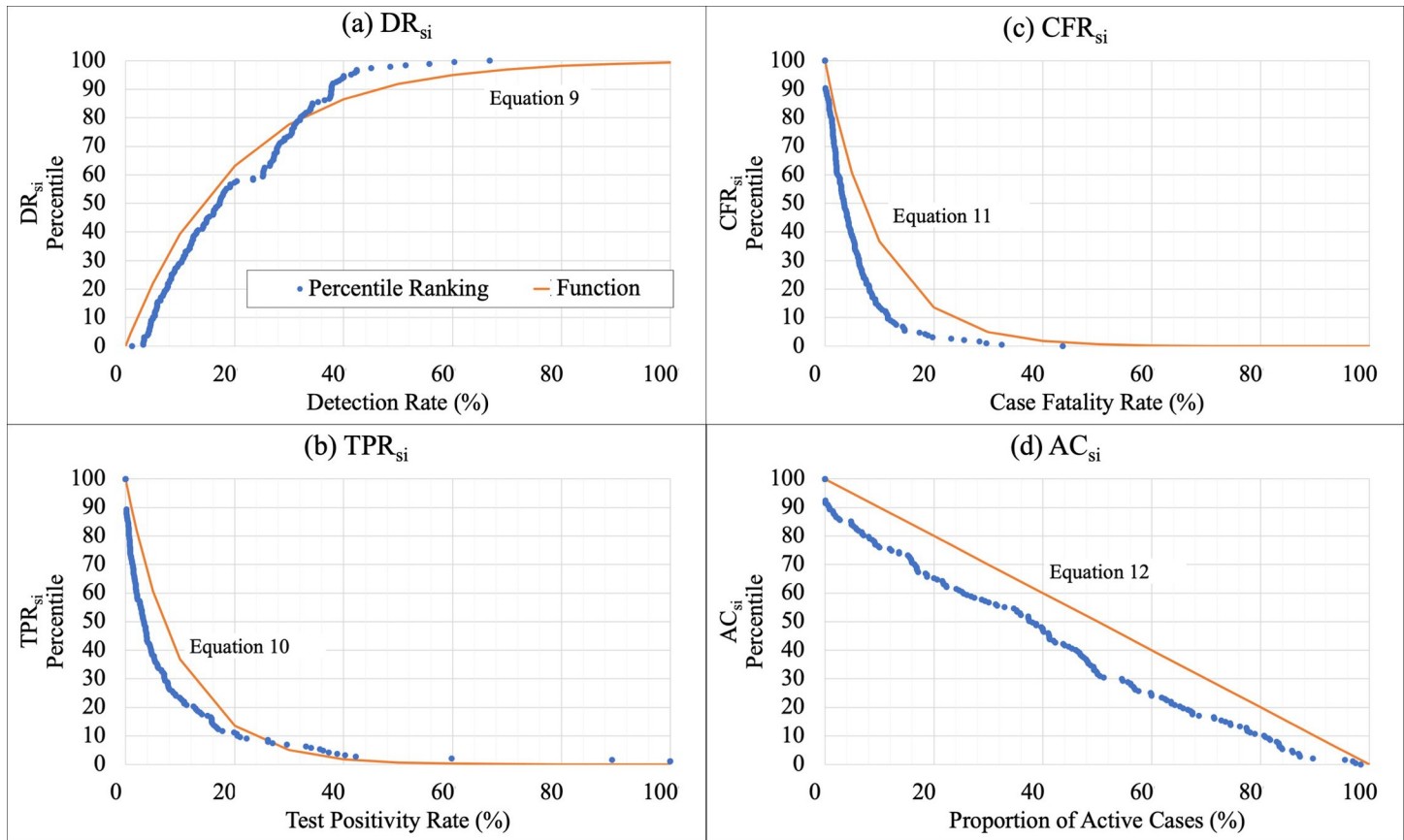

**Fig 2. Percentile rankings and sub-index functions for four key indicators.** Percentile rankings, in which 100 represented to most desirable value of the indicator and 0 the least desirable, were computed for each of the four key indicators (a) DR, (b) TPR, (c) CFR, and (d) AC. Functions were fitted to the percentile ranks, which were used to compute the sub-indices (a) DR$_{si}$, (b) TPR$_{si}$, (c) CFR$_{si}$, (d) AC$_{si}$ (Eqs 9–12).

100. This relationship was approximated with (Eq 9).

$$DR_{si} = 100(1 - e^{-5DR})$$ (9)

**Test Positivity sub-index (TP$_{si}$).** The TPR is a commonly used testing metric for COVID-19 [40] and has been recommended to guide decisions regarding introducing or relaxing public health measures [41]. A high TPR represents a reactive, rather than proactive, approach to testing. Ranking showed an exponential decay relationship (Fig 2B), which was incorporated into the TP$_{si}$ function (Eq 10). If TPR was not available, a dummy value of 20 was used. TP$_{si}$ was given 20% weighting in computing the CovTI.

$$TP_{si} = 100e^{-10TPR}$$ (10)

**Case-Fatality sub-index (CF$_{si}$).** The CFR should approach the IFR of 1 percent (or less) if testing is adequate to detect the majority of infections. If CFR is higher, it is likely only severely infected patients are being diagnosed. Ranking showed an exponential decay relationship (Fig 2C), which was used to define the CF$_{si}$ function (Eq 11). CF$_{si}$ was given 20% weighting. If the CFR was 0% (i.e., no deaths had yet been recorded as attributable to COVID-19), a dummy value of 50 was used.

$$CF_{si} = 100e^{-10CFR}$$ (11)

**Active Case sub-index (AC$_{si}$).** Finally, a fourth sub-index accounted for the activeness of the epidemic in a country. If an outbreak is active in a country, it is less likely the testing is adequate, with the increase possibly reflecting inadequate case identification. This sub-index also provides a metric to reflect progress as cases resolve. A lower AC is indicative that the epidemic is not increasing exponentially. Ranking showed a linear relationship (Fig 2D), which was used to develop the AC$_{si}$ (Eq 12). It was given a weight of 20%. In locations where AC was not computable, a dummy value of 50 was used.

$$AC_{si} = 100 - 100AC$$ (12)

**COVID-19 Testing Index (CovTI).** The CovTI was calculated as the weighted average of the four sub-indices (Eq 13) described above with a heavier weighting given to the DR$_{si}$ due to the importance of undetected cases in driving infections and because it most directly represents testing effectiveness (the primary objective of the index), whereas the other sub-indices are secondary indicators of testing effectiveness. CovTI was computed for the countries and territories meeting the inclusion criteria (n = 165).

$$CovTI = 0.4\,DR_{si} + 0.2\,TP_{si} + 0.2\,CF_{si} + 0.2\,AC_{si}$$ (13)

## Statistical analyses

Five independent grouping variables were assessed for their relationship with COVID-19 testing effectiveness by analyzing their association with CovTI (Table 1). Testing and contact tracing policy status were accessed from the Oxford COVID-19 Government Response Tracker [42] for May 13, 2020, which is three weeks prior to June 3, 2020, approximately the average time from symptom onset to death [22]. Islands were defined as any country that is an island, is part of an island (co-island), has limited land connections (limited land), or is an archipelago (see details in S1 Table). Any location for which at least one independent variable was not

**Table 1. Definitions of grouping variables for multiple linear regression of COVID-19 Testing Index (n = 147).**

| Factor (Grouping Variables) | Operational Definition | n (%) |
|---|---|---|
| **Testing Policy** | | |
| No testing policy | OxCGRT Testing = 0[a] | 6 (4.1) |
| Limited Testing | OxCGRT Testing = 1[a] | 60 (40.8) |
| Symptomatic Testing | OxCGRT Testing = 2 [a] | 53 (36.1) |
| Open Public Testing | OxCGRT Testing = 3 [a] | 28 (19.1) |
| **Contact Tracing Policy** | | |
| No contact tracing | OxCGRT Contact Tracing = 0 [a] | 18 (12.2) |
| Limited contact tracing | OxCGRT Contact Tracing = 1[a] | 58 (39.5) |
| Comprehensive contact tracing | OxCGRT Contact Tracing = 2 [a] | 71 (48.3) |
| **Geographical setting** | | |
| Island or island-like nation | Entirely on island(s) (including Australia) or parts of islands (e.g., Dominican Republic) or have very limited land connections (e.g., Hong Kong) | 23 (15.7) |
| Non-island nation | Not meeting the definition of island nation | 124 (84.3) |
| **Form of government** | | |
| Federation | Constitutionally a federation | 24 (16.3) |
| Unitary state | Constitutionally not a federation | 123 (83.7) |
| **Development status** | | |
| OECD[b] member | Member | 36 (24.5) |
| Non-OECD member | Not a member | 111 (75.5) |

[a]OxCGRT = Oxford COVID-19 Government Response Tracker [42] for May 13, 2020.
[b]OECD = Organization for Economic Development.

defined was excluded from analysis, creating an analysis data subset (n = 147). Crude bivariate analyses using two-tailed two-sample t-tests and one-way analysis of variance (ANOVA) were used to test whether the means between groups were different. A multiple linear regression (MLR) model was developed by using forced entry of all factors. Factors were removed by backwards stepwise method (p >0.05) with Bayesian Information Criterion (BIC) used to assess model fit and overparameterization. Analyses were performed in Stata 14 [43].

## Sensitivity analysis

We conducted a sensitivity analysis to assess how robust the model was in response to uncertainty in chosen model parameters. Eight scenarios were evaluated in the sensitivity analysis. Two scenarios (A1 and A2) assessed the cap on computed CFR (Eq 1); two scenarios (B1 and B2) assessed the assumed IFR (Eq 5); two scenarios (C1 and C2) assessed the $m_{demsys}$ multiplier (Eq 6); one scenario (D1) assessed the TPR threshold (Eq 7); and one scenario (E1) assessed the weighting of the sub-indices (Eq 13). The robustness of the model's conclusions was assessed in two different ways. The F-statistic and adjusted $R^2$ values from the MLR model using data computed for each scenario was compared to the final model's values. The model was considered robust and insensitive to the inherent assumptions if the F-statistic and adjusted $R^2$ were within 10% of the final model's values and if the interpretation of the results were unchanged.

## Results

### True number of infections and detection rate

Globally, the model estimated that approximately 65.7 million people have been infected with SARS-CoV-2 in the period prior to June 3, 2020, compared to the reported 6.47 million cases (mean multiplier factor, $f$, = 10.2, range = 1.5–85). In other words, for every reported case it was estimated that 9.5 infections had gone unreported in that period.

The global DR was estimated to be 9.8% (range = 1.2–66.8%) in the period prior to June 3, 2020. The countries estimated to have had the highest DRs over that time period were Australia (66.8%) and Iceland (60.3%) (See S1 Table for full results).

### Comparison to previous estimates

This model's estimates were compared against historical estimates in the literature (Table 2). The results showed that this model's estimates were similar to previous estimates at comparable time periods ($R^2$ = 0.44). In many cases the estimates of true number of infections and DR closely matched previous estimates, and in most cases the estimates were within the 95% confidence interval of previous estimates.

### COVID-19 Testing Index

**Comparison between countries.** Countries in the top quartile of CovTI had lower TPR, lower CFR, lower proportion of active cases, and higher DR compared to other quartiles (Fig 3). The top 15 countries according to the index included many island nations and states that are effectively islands (e.g., Hong Kong, Iceland, Australia) (Table 3). Southeast Asian countries including Cambodia, Vietnam, Malaysia, Brunei and Thailand also had high CovTI. Full results are reported in Supporting Information (S1 and S2 Tables).

**Table 2. Comparison of period prevalence and detection rate estimates with other similar estimates in literature [44, 45].**

| Date | Country | Other Estimates [44, 45] | | This Study's Estimates | |
|---|---|---|---|---|---|
| | | Estimated Period Prevalence [95% CI] | Estimated Detection Rate [95% CI] | Estimated Period Prevalence | Estimated Detection Rate |
| | | % of total population | % of infections | % of total population | % of infections |
| 14 April 2020 [44] | Australia | 0.04 [0.03–0.09] | 59 [28–90] | 0.03 | 88 |
| | Canada | 1.0 [0.67–2.2] | 6.7 [3.1–10] | 0.44 | 21 |
| | South Korea | 0.06 [0.04–0.13] | 35 [16–53] | 0.06 | 33 |
| | USA | 2.6 [1.7–5.6] | 6.8 [3.2–10] | 1.6 | 14 |
| 4 May 2020 [45] | Austria | 0.76 [0.59–0.98] | 23.4 [18–30] | 1.0 | 17 |
| | Belgium | 8.0 [6.1–11] | 5.4 [3.9–7.1] | 12.3 | 3.5 |
| | Denmark | 1.0 [0.81–1.4] | 17 [12–21] | 0.95 | 17 |
| | France | 3.4 [2.7–4.3] | 7.6 [6.0–9.6] | 5.9 | 4.3 |
| | Germany | 0.85 [0.66–1.1] | 23 [18–30] | 1.1 | 18 |
| | Italy | 4.6 [3.6–5.8] | 7.5 [6.0–9.6] | 8.0 | 4.3 |
| | Norway | 0.46 [0.34–0.61] | 32 [24–44] | 0.52 | 28 |
| | Spain | 5.5 [4.4–7.0] | 9.6 [7.5–12] | 7.7 | 6.8 |
| | Sweden | 3.7 [2.8–5.1] | 6.0 [4.4–8.0] | 3.0 | 7.4 |
| | Switzerland | 1.9 [1.5–2.4] | 18 [14–23] | 3.1 | 11 |
| | UK | 5.1 [4.0–6.5] | 5.3 [4.2–6.8] | 5.4 | 4.9 |

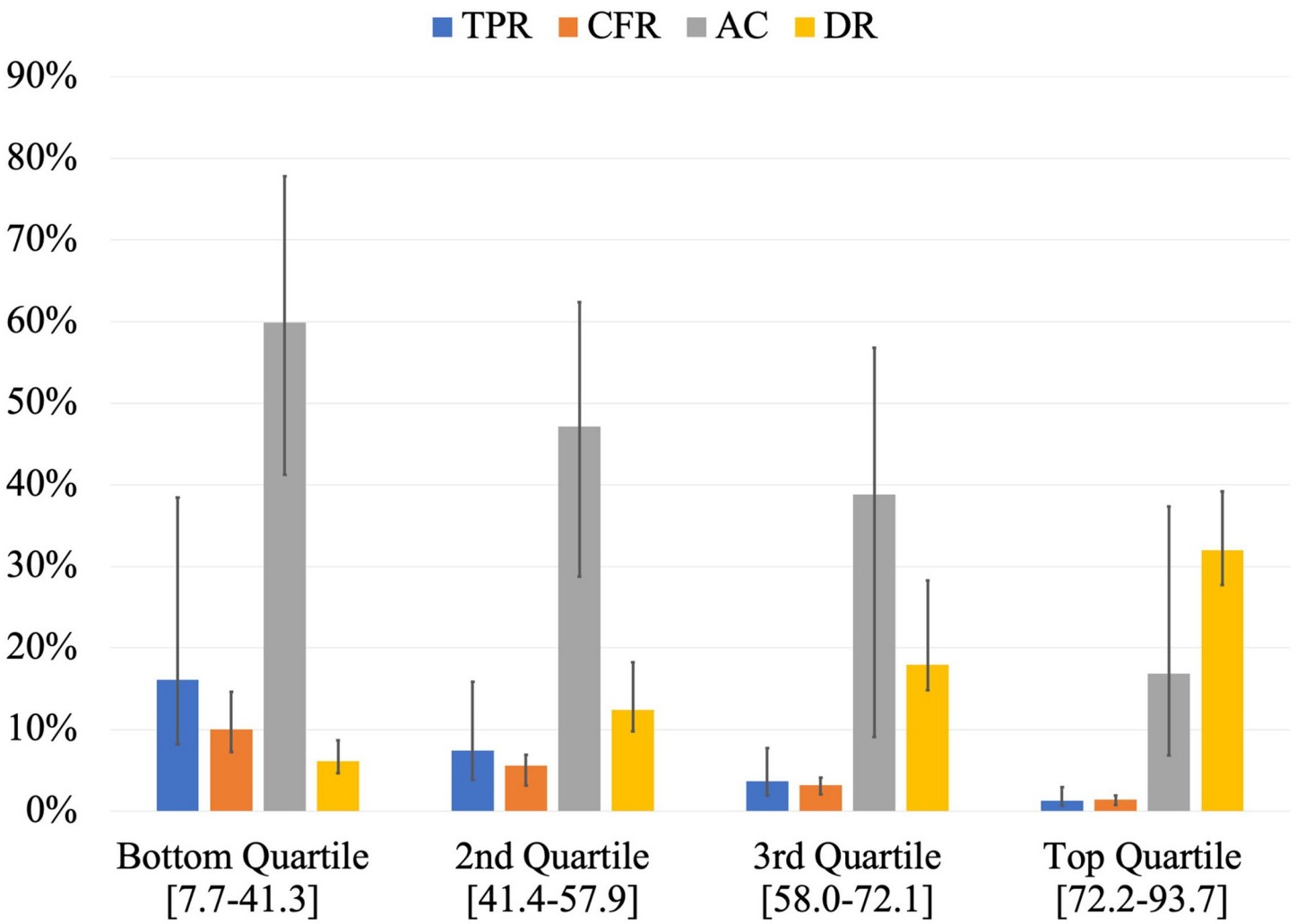

**Fig 3. Comparison of the medians of Test Positivity Rate (TPR), Case Fatality Rate (CFR), proportion of Active Cases (AC), and Detection Rate (DR) among the quartiles of COVID-19 Testing Index (CovTI).** Bottom quartile included the lowest 25% of CovTI values (n = 42), 2nd quartile included CovTI values between 25th and 50th percentiles (n = 41), 3rd quartile included CovTI values between 50th and 75th percentiles (n = 41), and the top quartile included highest CovTI values above 75th percentile (n = 41). Values in brackets indicate the range of CovTI values in each quartile. Error bars represent interquartile range. Data per 00:00 GMT June 3, 2020.

**Variable analysis.** Bivariate analyses showed that testing policy and contact tracing policy were significantly associated (p<0.0001) with CovTI (Table 4), with increasing levels of testing and contact tracing associated with higher CovTI. Islands had significantly higher CovTI than non-islands (p = 0.004). Unitary states had higher CovTI compared to federations, but the differences were not significant (p = 0.26). The CovTI values among OECD members and non-members were nearly equal.

All factors were entered into the initial MLR model. The final model included all factors except economic development (OECD member or non-member). The MLR showed that testing policy had the largest effect on testing outcomes, whereby widespread open testing was associated with a 31.1-point increase in CovTI compared to no testing policy (Table 5). Contact tracing, centralized governments, and islands were also associated with improved CovTI values. However, the difference in CovTI associated with type of government was not statistically significant (p = 0.20).

**Table 3. COVID-19 Testing Index (CovTI) and sub-indices a) among top 15 countries and territories assessed (n = 165).**

| Global Rank | Country or Territory | CovTI | $DR_{si}$ (40%) | $TP_{si}$ (20%) | $CF_{si}$ (20%) | $AC_{si}$ (20%) |
|---|---|---|---|---|---|---|
| 1 | Hong Kong | 93.7 | 91.2 | 94.8 | 96.2 | 95.2 |
| 2 | Australia | 93.5 | 96.5 | 95.3 | 85.9 | 93.2 |
| 3 | Iceland | 91.8 | 95.1 | 74.4 | 94.6 | 99.9 |
| 4 | Cambodia | 91.3 | 81.9 | 94.2 | 100.0 | 98.4 |
| 5 | New Zealand | 90.6 | 85.9 | 94.8 | 86.4 | 99.9 |
| 6 | Vietnam | 90.2 | 80.7 | 98.8 | 100.0 | 90.9 |
| 7 | Taiwan | 89.9 | 86.2 | 94.1 | 85.1 | 98.0 |
| 8 | Réunion | 85.9 | 84.9 | 75.8 | 97.6 | 86.4 |
| 9 | Malta | 85.7 | 80.3 | 91.6 | 85.2 | 90.8 |
| 10 | Palestine | 85.5 | 80.9 | 90.4 | 92.3 | 83.1 |
| 11 | Malaysia | 84.7 | 84.6 | 86.9 | 84.0 | 83.6 |
| 12 | Brunei | 84.5 | 71.8 | 93.1 | 86.7 | 99.3 |
| 13 | Thailand | 84.5 | 74.4 | 92.9 | 82.5 | 98.1 |
| 14 | South Korea | 83.0 | 78.2 | 88.4 | 77.6 | 92.9 |
| 15 | Rwanda | 82.9 | 78.2 | 94.7 | 92.9 | 70.6 |

$DR_{si}$, $TP_{si}$, $CF_{si}$, $AC_{si}$, as described in text with percentages indicating degree of weighting. Data per 00:00 GMT June 3, 2020. Complete dataset in S1 Table.

## Model robustness

The results of the sensitivity analysis showed that possible uncertainty in the model's assumptions did not affect the outcomes of the study. The scenarios that caused the largest change in the F-statistic were scenarios B1 (IFR of 0.5%) and E1 (equal weighting to each of the four

**Table 4. Comparison of differences in COVID-19 Testing Index between countries and states with different testing and tracing policies, geographical settings, forms of government and economic development status (n = 147).**

| Variable | n (%) | CovTI $\bar{x}$ (95% CI) | Test statistic[a] | p-value |
|---|---|---|---|---|
| **Testing policy** | | | 8.4 | <0.0001 |
| No testing policy | 6 (4.1) | 29.7 (17.6–41.9) | | |
| Limited testing | 60 (40.8) | 54.0 (49.0–59.0) | | |
| Symptomatic testing | 53 (36.1) | 59.6 (54.7–64.5) | | |
| Open public testing | 28 (19.1) | 66.9 (61.0–72.8) | | |
| **Contact tracing policy** | | | 9.8 | 0.0001 |
| No contact tracing | 18 (12.2) | 43.9 (35.4–52.5) | | |
| Limited contact tracing | 58 (39.5) | 54.4 (49.4–59.3) | | |
| Extensive contact tracing | 71 (48.3) | 63.5 (59.3–67.7) | | |
| **Geographical setting** | | | 2.94 | 0.004 |
| Island or island-like | 23 (15.7) | 68.0 (59.7–76.3) | | |
| Non-island | 124 (84.4) | 55.5 (52.5–58.8) | | |
| **Form of government** | | | 1.12 | 0.26 |
| Unitary state | 123 (83.7) | 58.3 (54.8–61.7) | | |
| Federation | 24 (16.3) | 53.5 (45.4–61.5) | | |
| **Economic development status** | | | 0.08 | 0.94 |
| OECD Member | 36 (24.5) | 57.3 (50.6–63.9) | | |
| Non-OECD Member | 111 (75.5) | 57.6 (54.0–61.2) | | |

[a] t-statistic from two-way two-sample t-test with equal variance or F-statistic from one-way ANOVA.

**Table 5. Multiple linear regression analysis F(7, 139) = 7.07 (p<0.0001), adjusted R² = 0.23 of factors associated with COVID-19 Testing Index from final model (n = 147) (data from 00:00 GMT June 3, 2020).**

| Independent variables | Mean CovTI (95% CI) | β coefficient (change in CovTI associated with factor) | 95% CI | p value |
|---|---|---|---|---|
| **Constant** | | 20.6 | 4.6–36.6 | |
| **Testing policy** | | | | |
| No testing policy | 29.7 (17.6–41.9) | Ref. | | <0.0001 |
| Limited testing | 54.0 (49.0–59.0) | 19.4 | 4.5–34.2 | |
| Symptomatic testing | 59.6 (54.7–64.5) | 24.3 | 9.4–39.2 | |
| Open public testing | 66.9 (61.0–72.8) | 31.1 | 15.2–47.0 | |
| **Contact tracing policy** | | | | 0.001 |
| No contact tracing | 43.9 (35.4–52.5) | Ref. | | |
| Limited contact tracing | 54.4 (49.4–59.3) | 6.1 | -3.3–15.4 | |
| Extensive contact tracing | 63.5 (59.3–67.7) | 12.4 | 3.1–21.7 | |
| **Geographical setting** | | | | 0.007 |
| Non-island | 55.5 (52.5–58.8) | Ref. | | |
| Island or island-like | 68.0 (59.7–76.3) | 10.6 | 2.9–18.3 | |
| **Form of government** | | | | 0.20 |
| Federation | 53.5 (45.4–61.5) | Ref. | | |
| Unitary state | 58.3 (54.8–61.7) | 5.1 | -2.6–12.8 | |

CovTI sub-indices). In all scenarios, however, the F-statistic and adjusted R² from the MLR were within 10% of the respective values for the final model. Additionally, the interpretation of the results (e.g., which factors were statistically significant) did not differ from the final model. Full results are reported in Supporting Information S1 Appendix.

## Discussion

We developed a novel comprehensive metric, entitled CovTI, that attempted to measure effectiveness of testing during the first four months of the COVID-19 pandemic at the country level and was derived from key epidemiological indicators computable from data available across nearly all countries, as reported on the Worldometer website [30]. Previous research has assessed government response [26] and suggested specific indicators to facilitate inter-country comparisons [27]. However, this is the first published metric to comprehensively assess testing outcomes with a focus on detection/underreporting.

### Key strengths and limitations

Mitigation efforts to contain the spread of SARS-CoV-2 should include testing, contact tracing, and isolation of cases, alongside social distancing and face masks [4, 5]. National policy decisions and individual choices must be informed by an assessment of risk that is supported by data [46]. CovTI aimed to provide an additional data point that holistically combined four important epidemiological indicators into a single metric. As such, it facilitated inter-country comparisons that elucidated extrinsic factors associated with improved testing outcomes.

Another advantage to using CovTI is that it incorporated a parsimonious empirical model to estimate the period prevalence and detection rate. Despite the parsimoniousness, the results were comparable to more complex modeling approaches [44, 45]. The findings suggested that nearly 90% of global infections were unreported in the first four months of the pandemic, which was consistent with previous estimates showing that true number of infections are many times higher than reported cases [10–12, 17]. In addition, the true number of infections as a percentage of the population has been modeled in various European countries. For example, period prevalence in

Italy was estimated at 4 percent in April [47] and 4.4 percent in France in May [48]. Several of the hardest affected countries early in the pandemic had an estimated period prevalence between 3 and 7.5 percent [49]. These estimates were generally consistent with our estimates.

Seroprevalence is another way to assess the period prevalence of COVID-19. Nationwide seroprevalence studies, such as one in Spain, provided period prevalence estimates (5.0% through May 11, 2020) consistent with this and other models [50]. However, several factors can substantially affect the accuracy of seroprevalence studies, including low specificity, cross-reactivity with other viruses, high false positive false positive rates in low prevalence environments, and study biases [51–54]. Therefore, models and parsimonious estimates may continue to play an important role in estimating the true number of infections.

CovTI has several limitations due to its inherent assumptions. In order to calculate DR, the model assumed specific and universal relationships between deaths and total number of infections, implying an inherent IFR. Advances in therapeutics and differences in health system capacity will influence this rate though [55]. In addition, several factors including age, sex, hypertension, diabetes, and blood groups are known to affect mortality and hospitalization rates [56–63]. These factors were not accounted for in this model.

Furthermore, death data, which contributed substantially to the computation of CovTI, is not equitably comparable across all countries. Excess all-cause mortality (i.e., total mortality in excess of seasonal averages) are substantially higher than the reported COVID-19 deaths in many locations, including Brazil, Jakarta, New York City, and Ecuador [64]. These excess deaths suggest such locations may not have accurately captured deaths caused by COVID-19 in official figures or are experiencing increased mortality related to the pandemic but not necessarily due to infections [65, 66]. Different definitions of attributable deaths substantially affect death data. For example, Russia had previously used a very limited definition for inclusion determined via autopsy that does not count many deaths even if the patient previously tested positive for SARS-CoV-2 [67]. On the other hand, some countries have opted to exhaustively include any presumptive death to COVID-19 in official data. Belgium has included unconfirmed deaths within their COVID-19 death total [68]. In Belgium, COVID-19 deaths were greater than excess deaths [64]. These examples of limited or conservative death definitions impact this model's estimates. Adjusting death data based on excess mortality data could improve the the model; however, such data limited [69].

The model also assumed specific relationships between proxy indicators, such as the Global Health Security Index and Democracy Index, and data outcomes. While regression showed they were associated with the level of underreporting, these factors are indirect proxies for a more complex set of variables that determine underreporting. Furthermore, this model aimed to be parsimonious (i.e., not introducing excessive parameters or uncertainty) and is, by nature, deterministic. The decision to include a minimum number of variables and data was strategic and aimed to avoid overparameterization, but a stochastic approach could better illustrate the uncertainty and sensitivity to the above assumptions. The model also reported estimates of total number of cases and detection rates. These values should be used cautiously as a comparative tool, rather than exact values, alongside other indicators.

## Policy implications

High CovTI values were found in countries that have been recognized for their success in responding to the COVID-19 pandemic, including New Zealand, Taiwan, Australia, Iceland, and South Korea. These countries have had high testing rates, comprehensive contact tracing programs, and relative success in mitigating the health impacts of the virus [70–73]. Thus, this analysis provided quantitative evidence supporting policy recommendations to facilitate

strong national leadership, expand diagnostic capacity, rapidly enact comprehensive contact tracing, and proactively test for SARS-CoV-2 [70, 71].

Further supporting this notion, aggressive contact tracing and inclusive testing policies were found to be independently associated with increased testing effectiveness, as indicated by CovTI. This result both validates the model's ability to track COVID-19 testing outcomes and provides further confirmation that countries that prioritize policies and dedicate resources specifically to testing and contact tracing have effectively reduced underreporting and improved testing-related health outcome metrics. Previous studies have found an association between expansive testing early in the pandemic and improved mortality data in Germany, South Korea, and Iceland [74]. On the other hand, socioeconomic status had no significant association to the outcome in this study. Previous research has shown that policy decisions are more important than socioeconomic status [75].

Increased testing capacity is not a panacea, though. More testing is not necessarily better, especially if accuracy is overestimated or the testing is poorly targeted [76]. In the first four months of the pandemic, COVID-19 diagnoses were routinely confirmed through testing that employed the reverse transcriptase polymerase chain reaction (RT-PCR) technique [77]. The heavy reliance on this method has been questioned [78]. Alternative molecular detection techniques may be needed, especially if they can return results more quickly [79].

Interestingly, this research found a significant relationship between island nations and higher CovTI. Geographical isolation is an obvious advantage to controlling infectious disease [80]. Many island nations have chosen a strategy of eradication [81]. Geographic isolation and easier enactment of border closures have benefited island nations in responding to the pandemic [82].

Finally, it should be noted that while the estimates from this model showed substantial underreporting of infections, the overall period prevalence remained less than 10 percent in nearly all countries. Such a low proportion of the population presumably with antibodies is far from conferring herd immunity that may inhibit future disease transmission. It is important to note that, while these proportions are much higher than the officially reported cases, they do not represent herd immunity—a concept considered important to fully reopening society. Although herd immunity depends on the effective reproductive number ($R_e$) [83], which varies with effectiveness of interventions, some estimates specify a threshold of 50 to 60 percent seroprevalence to achieve herd immunity [84], while others, accounting for differential susceptibility, estimate the threshold may be as low as 20 percent [85]. Nevertheless, these estimates suggest that herd immunity has not yet occurred in the first four months of the pandemic at the national level of the countries analyzed. Therefore, achieving herd immunity through natural infection may be costly or unachievable [86].

## Future research

We encourage other researchers to build on this analysis by combining this metric with other databases that can account for other possible factors, such as trust in government institutions, demographics, or urban/rural distribution. Such analyses could further elucidate other extrinsic factors related to COVID-19 testing outcomes. Research could also be done to understand how association with other factors changes over time, as the pandemic progresses through different stages. Sub-national analyses may also be possible using the mathematical relationships defined in this index.

## Conclusion

This report described a novel comprehensive metric (COVID-19 Testing Index, CovTI) that evaluates the overall effectiveness of COVID-19 testing in the current pandemic using real-

time publicly reported data among 165 countries and territories. The metric incorporated case-fatality rate, test positivity rate, proportion of active cases, and an estimate of detection rate based upon reported death data by adjusting for heterogeneity in testing levels, health system capacity, and government transparency. The estimated detection rate of COVID-19 aligned satisfactorily with previous empirical and epidemiological models. National policies that facilitated open public testing and extensive contact tracing were significantly associated with higher values of CovTI, which reflected improvements in the estimated detection rate. Extrinsic factors, including geographic isolation and centralized forms of government, were also shown to be associated with improved COVID-19 testing outcomes. Countries should commit to expanding policies on testing and contact tracing in order to reduce levels of undetected infections and reduce disease transmission. Applications of this metric include combining it with different databases to identify other factors that affect testing outcomes or using it to temporally track a holistic measure of testing outcomes at the national or sub-national level.

## Supporting information

**S1 Table. Raw data inputs and computed values for the COVID-19 Testing Index (CovTI) on June 3, 2020.** Raw data inputs, key epidemiological indicators, multipliers, factors, and sub-indices used to compute CovTI on June 3, 2020 among included countries and territories (n = 165). Further details of each variable are described in the text. C = cases, D = total deaths, R = total recovered, A = active cases, P = population (in millions), T = total tests, CFR = case fatality rate, TPR = test positivity rate, $m_{demsys}$ = multiplier to account for health system capacity and democratic transparency, $f_1$ = factor 1, $f_2$ = factor 2, Inf = true number of infections, Prev = Estimated Period Prevalence = Inf /P, Act = proportion active cases = A/C, DR = detection rate, DRsi = Detection Rate sub-index, TPsi = Test Positivity sub-index, CFsi = Case-Fatality sub-index ACsi = Active Case sub-index, CovTI = COVID-19 Testing Index. OECD = Organization for Economic Development member, BRIC = Brazil, Russia, India, and China.
(DOCX)

**S2 Table. Dataset for bivariate and multiple linear regression analyses.** CovTI as of June 3, 2020; island status; form of government; OECD membership; COVID-19 testing policy as of May 13, 2020; and COVID-19 contact tracing policy as of May 13, 2020 among included countries and territories with complete data (n = 147). Further details of each variable are described in the text.
(DOCX)

**S1 Appendix. Sensitivity analysis using alternative scenarios of model construction.** Eight alternative scenarios were considered. In each scenario, CovTI (and all subsequent inputs) was computed. Then, the MLR was run. MLR results were compared to the final model's results.
(DOCX)

## Author Contributions

**Conceptualization:** Anthony C. Kuster.

**Data curation:** Anthony C. Kuster.

**Formal analysis:** Anthony C. Kuster.

**Funding acquisition:** Hans J. Overgaard.

**Methodology:** Anthony C. Kuster.

**Software:** Anthony C. Kuster.

**Supervision:** Hans J. Overgaard.

**Validation:** Hans J. Overgaard.

**Visualization:** Anthony C. Kuster, Hans J. Overgaard.

**Writing – original draft:** Anthony C. Kuster, Hans J. Overgaard.

**Writing – review & editing:** Anthony C. Kuster, Hans J. Overgaard.

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
