## [Decision Letter · Decision Letter 0]

8 Oct 2020

A novel comprehensive metric to assess COVID-19 testing outcomes: Effects of geography, government, and policy response

PONE-D-20-20527

Dear Dr. Overgaard,

We’re pleased to inform you that your manuscript has been judged scientifically suitable for publication and will be formally accepted for publication once it meets all outstanding technical requirements.

Kind regards,

Karol Sestak

Academic Editor

PLOS ONE

2.We note that the figures  in your submission contain [map/satellite] images which may be copyrighted. All PLOS content is published under the Creative Commons Attribution License (CC BY 4.0), which means that the manuscript, images, and Supporting Information files will be freely available online, and any third party is permitted to access, download, copy, distribute, and use these materials in any way, even commercially, with proper attribution. For these reasons, we cannot publish previously copyrighted maps or satellite images created using proprietary data, such as Google software (Google Maps, Street View, and Earth). For more information, see our copyright guidelines: http://journals.plos.org/plosone/s/licenses-and-copyright.

1.    You may seek permission from the original copyright holder of the figures to publish the content specifically under the CC BY 4.0 license. 

3.We note that you have indicated that data from this study are available upon request. PLOS only allows data to be available upon request if there are legal or ethical restrictions on sharing data publicly. For information on unacceptable data access restrictions, please see http://journals.plos.org/plosone/s/data-availability#loc-unacceptable-data-access-restrictions.

Reviewers' comments:

Reviewer's Responses to Questions

**Comments to the Author**

1. Is the manuscript technically sound, and do the data support the conclusions?

Reviewer #1: Yes

2. Has the statistical analysis been performed appropriately and rigorously? 

Reviewer #1: Yes

3. Have the authors made all data underlying the findings in their manuscript fully available?

Reviewer #1: Yes

4. Is the manuscript presented in an intelligible fashion and written in standard English?

Reviewer #1: Yes

5. Review Comments to the Author

Reviewer #1: - The manuscript has already been submitted as preprint in medRxiv.

- It describes the new metric (Testing Index = TI) that could be useful along with other, already established metrics, in evaluating and comparing the COVID-19 infection burdens among different countries/regions.

- One of the important justifications for using the TI is that this metric incorporates relative or proportional type of measurements. It is often stated in these days by politicians and non-experts that “increased testing is counterproductive because it inevitably reveals increased number of cases….”. Relative or proportional types of metrics by their nature eliminate such criticism and at the same time should be used and explained to public in order to corroborate absolute data.

- Authors emphasize the need of a single comprehensive metric that could be used universally as it would encompass characteristics of both absolute and relative metrics.

- The results presented here are statistically sound and strongly suggest that COVID-19-TI could be used for such a purpose.

6. PLOS authors have the option to publish the peer review history of their article (what does this mean?). If published, this will include your full peer review and any attached files.

Reviewer #1: **Yes: **Dr. Karol Sestak

---

## [Author Response · Author response to Decision Letter 0]

9 Nov 2020

Please see uploaded response to reviewer's comments.

---

## [Decision Letter · Decision Letter 1]

19 Jan 2021

PONE-D-20-20527R1

A novel comprehensive metric to assess COVID-19 testing outcomes: Effects of geography, government, and policy response

PLOS ONE

Dear Dr. Overgaard,

Thank you for submitting your manuscript to PLOS ONE. After careful consideration, we feel that it has merit but does not fully meet PLOS ONE’s publication criteria as it currently stands. Therefore, we invite you to submit a revised version of the manuscript that addresses the points raised during the review process.

Your manuscript received mixed reviews in the first round of revision; subsequently, Reviewer 2, who raised a number of concerns, was unavailable in the next round to assess the revisions to your manuscript. As such, an additional reviewer was invited for this round of revision, who raised some new and important concerns regarding your study. Specifically, they noted the following:

1-  The scientific content of the paper lacks rigour : the authors take into account mostly stochastic factors (such as the political measures concerning the tests which only changed overnight depending on the availability of medical means, especially during the first phase of the covid) to ultimately propose a deterministic indicator without any analysis of the uncertainties linked to the stochasticity of the input data.

2- Some relationships have been considered linear without any justification and sometimes wrongly, for example, why is the real number of infected people linearly proportional to the number of reported cases?

The reviewer’s full comments can be viewed in full below. Please note that further consideration is dependent on the submission of a manuscript that addresses all the concerns raised in this round of review.”

We look forward to receiving your revised manuscript.

Kind regards,

Laurent Pujo-Menjouet

Academic Editor

PLOS ONE

Reviewers' comments:

Reviewer's Responses to Questions

**Comments to the Author**

1. If the authors have adequately addressed your comments raised in a previous round of review and you feel that this manuscript is now acceptable for publication, you may indicate that here to bypass the “Comments to the Author” section, enter your conflict of interest statement in the “Confidential to Editor” section, and submit your "Accept" recommendation.

Reviewer #3: (No Response)

2. Is the manuscript technically sound, and do the data support the conclusions?

Reviewer #3: No

3. Has the statistical analysis been performed appropriately and rigorously? 

Reviewer #3: No

4. Have the authors made all data underlying the findings in their manuscript fully available?

Reviewer #3: Yes

5. Is the manuscript presented in an intelligible fashion and written in standard English?

Reviewer #3: Yes

6. Review Comments to the Author

Reviewer #3: A novel comprehensive metric to assess COVID-19 testing outcomes: Effects of geography, government, and policy response.

Referee: #3

This paper develops a metric to assess COVID-19 testing outcomes ex- pressed as a linear combination of the rates CFsi, TPsi, ACsi and DRsi. These rates are computed with adjusted reported death data, the health system capacity and level of democracy of the countries.

The idea of setting up a metric to compare the level of effectiveness of COVID tests in different countries is of major interest. However, the scientific content of the paper lacks rigor and seems to me very far from the framework and the level of requirement of the journal PLOS. In fact, the authors take into account mostly stochastic factors (such as the political measures concerning the tests which only changed overnight depending on the availability of medical means, especially during the first phase of the covid) to ultimately propose a deterministic indicator without any analysis of the uncertainties linked to the stochasticity of the input data.

Some relationships have been considered linear without any justification and sometimes wrongly, for example, why is the real number of infected people linearly proportional to the number of reported cases?

I agree to the great interest to have a good metric to assess COVID-19 testing outcomes however, the one proposed in this paper seems not relevant and not rigorous. For all these reasons and despite the answers given to the first phase of the review, I will not recommend the publication of this result in PLOS ONE journal.

7. PLOS authors have the option to publish the peer review history of their article (what does this mean?). If published, this will include your full peer review and any attached files.

Reviewer #3: No

---

## [Author Response · Author response to Decision Letter 1]

2 Feb 2021

Please see our detailed response in the uploaded Response to Reviewers document.

---

## [Editor Report · Decision Letter 2]

22 Feb 2021

A novel comprehensive metric to assess effectiveness of COVID-19 testing: Inter-country comparison and association with geography, government, and policy response

PONE-D-20-20527R2

Dear Dr. Overgaard,

We’re pleased to inform you that your manuscript has been judged scientifically suitable for publication and will be formally accepted for publication once it meets all outstanding technical requirements.

Kind regards,

Laurent Pujo-Menjouet

Academic Editor

PLOS ONE

Additional Editor Comments (optional):

After reading the comments of the readers on the second revised version, I consider at that point that they answered all the points and can be accepted for publication.

Best,

Laurent
---

## [Editor Report · Acceptance letter]

24 Feb 2021

PONE-D-20-20527R2 

A novel comprehensive metric to assess effectiveness of COVID-19 testing: Inter-country comparison and association with geography, government, and policy response 

Dear Dr. Overgaard:

I'm pleased to inform you that your manuscript has been deemed suitable for publication in PLOS ONE. Congratulations! Your manuscript is now with our production department. 

Kind regards, 

on behalf of

Dr. Laurent Pujo-Menjouet 

Academic Editor

PLOS ONE